# Extranodal Marginal Zone Lymphoma: Pathogenesis, Diagnosis and Treatment

**DOI:** 10.3390/cancers14071742

**Published:** 2022-03-29

**Authors:** Alice Di Rocco, Luigi Petrucci, Giovanni Manfredi Assanto, Maurizio Martelli, Alessandro Pulsoni

**Affiliations:** Hematology, Departement of Traslational and Precision Medicine, Sapienza University of Rome, 00161 Rome, Italy; dirocco@bce.uniroma1.it (A.D.R.); l.petrucci@bce.uniroma1.it (L.P.); assanto@bce.uniroma1.it (G.M.A.); martelli@bce.uniroma1.it (M.M.)

**Keywords:** non-Hodgkin lymphoma, marginal zone lymphoma, diagnosis, prognosis, treatment, immunotherapy, targeted-therapy, MALT, BALT, OAL

## Abstract

**Simple Summary:**

Extranodal marginal zone lymphoma (EMZL) is an indolent lymphoproliferative disease morphologically composed of small heterogeneous B lymphocytes. It generally occurs with a localized stage and can arise in various organs, the most frequent being the stomach, lung, and ocular adnexa. Depending on the presentation and the possible association with infectious agents, different therapeutic approaches are to be undertaken. The purpose of this review is to describe the biology underlying this pathology, the diagnostic, and therapeutic approach.

**Abstract:**

Extranodal Marginal Zone Lymphoma (EMZL lymphoma) is an indolent B-cell lymphoma with a median age at diagnosis of about 60 years. It accounts for 7–8% of all B-cell lymphomas. It can occur in various extranodal sites, including stomach, lung, ocular adnexa, and skin; furthermore, the disseminated disease can be found in 25–50% of cases. Several infectious agents, such as *Helicobacter pylori* (*H. Pylori*) in the case of gastric Mucosa Associated Lymphoid Tissue (MALT) Lymphoma, can drive the pathogenesis of this cancer, through the autoantigenic stimulation of T cells, but there may also be other factors participating such autoimmune diseases. Initial staging should include total body computed tomography, bone marrow aspirate, and endoscopic investigation if indicated. Fluorescence in situ hybridization (FISH), should be performed to detect the presence of specific chromosomal translocations involving the *MALT1* and *BCL10* genes, which leads to the activation of the *NF-κB* signaling pathway. Depending on the location and dissemination of the disease, different therapeutic choices may include targeted therapy against the etiopathogenetic agent, radiotherapy, immunochemotherapy, and biological drugs. The purpose of this review is to illustrate the complex biology and the diagnosis of this disease and to better define new treatment strategies.

## 1. Introduction

Mucosa-associated lymphoid tissue lymphoma (MALT) is a subtype of marginal zone non-Hodgkin’s lymphoma (MZL). This type of lymphoma can also be referred to as extranodal marginal zone lymphoma (EMZL) and can affect any mucosa.

EMZL is a distinct entity from other types of MZL: it differs from splenic marginal zone lymphoma (SMZL) and nodal marginal zone lymphoma (NMZL) in diagnostic criteria, genetic alterations, clinical behavior and therapy [1]. This review article summarizes the previously published literature on EMZL, with an overview of its clinical features, treatment options, and prognostic outcome.

## 2. Epidemiology

EMZL accounts for 50–70% of marginal zone lymphomas and 7–8% of non-Hodgkin’s lymphomas with a median age greater than 60 years at diagnosis. The most affected sites are the stomach, ocular adnexa, lung, and salivary glands. In the United States, the incidence is 18.3 cases per 1 million person-years [2].

### Etiopathogenesis

B-cells constituting marginal zone lymphoma originate from post-germinal center cells that usually can be found in the peripheral zone of the germinal center [1]. In most cases, rearrangement of the variable region of the immunoglobulin heavy chain can be detected, together with class switching and somatic mutations of light chain genes [3]. In the landscape of B-Cells lymphoma, the pathogenesis of EMZLs represents a perfect example of synergy between antigen chronic stimulation and genetic alterations that bring to lymphoma progression of B-Cell clones [4].

The occurrence of EMZL is, in almost all cases, associated with immune cross-reactions driven by chronic exposure to bacterial, viral, or autoimmune stimuli [5].

Genetic aberrations should follow the antigen/inflammation exposure, in a multi-step process, which can lead to tumorigenesis. Often B-cells exposed expand in specific clones, which end bearing acquired mutations. These lesions often result in the *NF-κB* pathway activation or *NOTCH* deregulation, which sustain lymphoma growth and reduce B-Cells dependence from chronic stimulation [6]. Nevertheless, a considerable portion of EMZL is responsive to the eradication of the etiological agent. *Helicobacter pylori* (*H. pylori*) can be found in 85% of Gastric EMZL and *Chlamydophila psittaci* (*C. psittaci*) in 10 to 50% of ocular adnexa EMZL; these are the most representative associations. In addition, Sjogren syndrome (Ss), Hepatitis C (HCV), and chronic sialadenitis are associated with more than 70% of salivary gland EMZLs [7,8] (Table 1). Once the underlying infection or autoimmune disorder is detected, specific treatment contributes to tumor remission although antibiotic therapy may have limits in its anti-lymphoma potential in extragastric mucosal sites [9]. The occurrence of genetic aberration is crucial for lymphomagenesis. Chromosomal aberrations, such as trisomy of chromosomes 3, 12, or 18, are detected in 20 to 30% of EMZL [10,11]. Moreover, chromosomal translocations are crucial in the etiopathology of disease and responsible for the deregulation of the NF-κB pathway. Translocations such as t(11;18)(q21;q21)/*BIRC3-MALT1* (formal API2-MALT1, present in 35% of non-Gastric EMZL; t(1;14)(p22;q32)/*IGH-BCL10*, present in 5%; t(14;18)(q32;q21)/*IGH-MALT1*, found in 15% of cases; t(3;14)(p14;q32)/IGHFOXP1, recently described are associated with antibiotic-resistant gastric lymphoma [10,11,12]. These translocations bring to altered levels of *MALT1* and *BCL10* proteins that lead to activation of the *NF-κB* pathway [13]. *NF-κB* is constituted by transcription factors, which interact into two signaling pathways (canonical and alternative) which in healthy subjects regulate also the transcription activation in response to extracellular stimuli (Figure 1) [14]. *BCL10* also regulates the non-canonical *NF-κB*. In the translocation t(14;18)(q32;q21), increased levels of paracaspase *MALT1*, an Arg-specific protease, work as an adjuvant to the interaction with *BCL10*, which triggers canonical *NF-κB* signaling and promotes the cleavage of negative regulators of *NF-kB* [15,16]. Mutations and/or deletions of *TNFAIP3* also can be detected in EMZLs resulting as well in upregulation of these pathways.

When present, *FOXP1* overexpression enhances WNT/β-catenin signaling and deregulates the *NF-κB* pathway [15]. Other occasional translocations have been reported involving *BCL6* and IGH genes. Gain of 6p25 is detected in approximately 20% of ocular adnexa EZML. In addition, somatic mutations can occur, especially targeting 5′ regulatory regions and coding sequences of proto-oncogenes (85% of gastric EMZL), somatic mutation could be due to abnormal target activation-induced cytidine deaminase (AID) in the germinal center. Missense mutations charging *PIM1* and *MYC* are described in 35% of EMZL and gain of function of *BLC10*, *MYD88*, and *NOTCH* ranging from 5 to 10% of cases [16]. In rare cases, the mutational process can end in aggressive lymphoma transformation.

## 3. Clinical Presentation and Disease Assessment

The clinical presentation of EMZL depends on the organ involved. Therefore, the symptoms are very different between the various subtypes: nodules and papules on the skin for cutaneous EMZL, recurrent respiratory infections are typical in lung EMZL, occult bleeding and dyspepsia are characteristic in gastric EMZL, while red eyes, epiphora or visual field defects are frequent in ocular EMZL [17]. EMZL may present with non-specific symptoms, such as low-grade fever, night sweats, malaise, abdominal pain, and weight loss in less than 5% of cases. Some symptoms may be related to the site involved. For respiratory EMZL, recurrent respiratory infections and/or discovery of lung nodules on imaging studies can be observed while for ocular adnexa, eye redness or slow-growing mass.

For an accurate diagnosis of EMZL, a biopsy of the affected site is necessary, and morphological, flow-cytometry, and genetic analysis are recommended. The role of Positron Emission Computed Tomography (PET-CT) is still controversial: there is a variable FDG avidity, which is higher in non-gastric lesions [18]. Morphological features include diffusely infiltrating cells associated with follicles that appear reactive. On immunophenotypic analysis, the cells are positive for B-cell markers CD19, CD20, and CD22 and negative for CD5, CD10, and CD23 [1].

## 4. Prognostic Factors and Disease Features

The MALT International Prognostic Index (IPI) can be used as a prognostic index, based on three variables: age over 70, stage III and IV, and elevated LDH. Therefore, three groups are defined: low, intermediate, and high with 5-year EFS equal to 70%, 56%, and 29%, respectively [19].

### 4.1. EMZL of Gastrointestinal Tract or Gastric MALT

EMZL involving the gastrointestinal tract represents around 50% of mucosa-associated lymphoid tissue (MALT-lymphoma). Gastric MALT accounts for 34% of the sites of origin of all EMZL, while the small intestine stands for 5 to 8%, the diffusion of *H. pylori* eradication leads to an incidence decrease of gastric MALT [20]. As it has always been considered, the ideal model of mucosa-associated lymphoma was initially recognized as a separate entity from other gastrointestinal B-Cell lymphomas because of its similarity in the histologic pattern architecture to gastric mucosa instead of lymph-node. The model in which chronic antigenic exposure and inflammation sustain the development of the disease is widely recognized [6,21]. Given the sterile gastric environment, the association between disease occurrence and *H. pylori* infection was rapidly evidenced [22]. T-regulatory cells and cytokines, especially including the TNF family, are involved in the inflammation sustaining tumorigenesis together with acquired mutations as reported above [23]. Chronic stimulation caused by *H. pylori* infection represents, in most cases, the event that triggers tumorigenesis. Cases of *H. pylori-negative* patients are anyway reported confirming that it is not the only stimulation responsible for the disease. *Campylobacter jejuni* (*C. jejuni*) infection was associated with small intestine EMZL [1,4,5]. The origin cell is still considered a post germinal center, deriving from B lymphocytes expressing clonal immunoglobulin (IgG or IgM). Sometimes immunoglobulins fragments are detectable in the bloodstream causing false-positive immunofixation [23]. At immunohistochemistry, gastric malt shows centrocyte-like cells and plasma cell differentiation with common B-surface markers, such as CD79, CD21, and CD20, but CD23, CD10, and Cyclin D1 are negative, and there are no specific markers expressed; CD5 is commonly negative with few CD5+ cases reported. Gastric MALT presenting a high percentage of centroblasts, initially recognized as “high-grade MALT lymphoma”, is nowadays considered as DLBCL subtype, occurring as composite or transformed EMZL. MALT cells share some features with plasma cells (CD20−) and often harbor the same translocation t(11;18)(q21;q21) [24]. Around 25 to 30% of gastric MALTs are estimated to harbor the *BIRC3-MALT1* translocation, which guarantees a survival advantage to tumor cells. Other genetic lesions mentioned above, such as t(14;18)(q32;q21)/immunoglobulin heavy locus (IGH)-*MALT1* and t(1;14)(p22;q32)/BCL10-IGH, can be found in *BIRC3/MALT1* negative patients [11,13,14]. Most mutations described are in any way involved in the NF-KB pathway, which constitutes a central role in the pathogenic mechanism. Integrins expression of gastric MALT lymphoma is higher than in other sites, conferring a greater tendency to home in on other mucosal sites [13]. For this reason, proper staging is crucial, and clinical and subclinical dissemination is present in 25–30% of patients [25].

Hyeon et al. in a population of 19 cases showed that the main genetic alterations included the genes involved in the activation pathway of the nuclear factor *NF-κB*: in 39% of TRAF3 a rearrangement of MALT1 was found, somatic mutations in 21% of *TRAF3*, in 16% the mutations of *TNFAIP3* (16%), and 16% those of *NOTCH1* (16%). In the *MALT1* rearrangement negative group, *TRAF3* mutations (33%), *TNFAIP3* somatic mutations (25%), and *NOTCH1* (25%) were among the most common alterations [26].

Gastric MALT is an indolent disease, and its symptoms are vague, the most common being dyspepsia, together with nausea and vomiting, and epigastric pain is frequent. Iron deficiency anemia and/or weight loss caused by pyloric stenosis can occur. Massive bleeding and perforation are rare complications at diagnosis [27]. Intestinal obstruction can occur mainly in the bowels involving disease. B-symptoms can occur, but a higher-grade lymphoma must be excluded. Mucosal biopsy performed during the endoscopic procedure is the gold standard for diagnosis. In mucosal tissues, lymphoepithelial lesions should be researched. The clonal marginal zone B cells typically infiltrate the epithelium, forming the spots that are considered a hallmark of MALT lymphoma, but this is, however, not indispensable for diagnosis [9,17,27]. Lymphoid monoclonality must be tested to rule out the diagnosis of inflammation and confirm lymphoma. Staging must include imaging, endoscopy procedures to detect disease in the gastrointestinal tract, ear/nose/throat examination, and ultrasound assessment of the lymph nodes. Endoscopic ultrasound should be performed to assess gastric wall infiltration and/or perigastric lymph nodes. Research for *H. pylori* (histochemistry, serology, fecal antigen, and breath test) and search for *MALT1* translocation by FISH are recommended. Bone Marrow biopsy is not mandatory in absence of clinical suspicion given the very low incidence of extra mucosal localization [9]. The role of (18F-FDG)-PET/CT is debated. In fact, only 50 to 60% with confirmed Gastric Lymphoma have positive PET; however, it is recommended in suspicion of nodal lesions or DLBCL transformation. The application of gallium-68 (Ga68)-labeled pentixafor for PET/MR targeting CXCR4 has been investigated showing promising results. MALT-IPI can be employed for risk stratification [18,19].

### 4.2. Bronchus-Associated Lymphoid Tissue (BALT) Lymphoma

BALT is found in all respiratory tracts, usually at the distal bronchial and bronchiolar districts, and is mainly composed of B cells with peripheral groups of T cells [28,29]. In adulthood, chronic antigenic stimulation can lead to neoplastic transformation of BALT and so to malignant lymphomas. It is the most common lung lymphoma, accounting for 70 to 90% of all cases [30]. The lung is the fourth most frequent site of EMZL occurrence in the United States after stomach, spleen, and eye with an incidence rate of up to 7.7%. The age of onset is between 60 and 70 years, with a slight predominance of the female sex [31]. The most frequent symptoms are fever, fatigue, cough, dyspnea, less frequent hemoptysis, and chest pain; fever and weight loss should lead to suspicion of a possible transformation into an aggressive form. About 30% of BALT lymphoma patients are asymptomatic [32].

Although infectious and inflammatory processes are frequently present in this type of lymphoma, unlike gastric MALT, an infectious agent capable of playing a certain role in the pathogenesis was not identified in the pulmonary EMZL. Adam et al. found the presence of the bacterium *Achromobacter xylosoxidans* in patients with pulmonary EMZL; however, this was not confirmed by a subsequent Japanese study [31,32]. Other studies have described possible correlations with Mycobacterium tuberculosis in patients not adequately treated with anti-tuberculosis drugs and other bacteria such as *Chlamydophila pneumonia*, *Chlamydophila trachomatis*, *C. psittaci*, and *Mycoplasma pneumonia* [33,34,35,36]. Therefore, there is not a certain correlation so far between infections and the onset of EMZL.

Pulmonary EMZL generally presents macroscopically a homogeneous, non-encapsulated tissue, with a color that can vary from white to light brown. Usually, the respiratory lumens are not involved, while the involvement of the visceral pleura can often be found. Optical microscopy shows a pulmonary architecture subverted by lymphoid infiltrates. The classic histological ‘triad’ of BALT lymphoma includes reactive lymphoid follicles, diffuse infiltration by centrocytes, and lymphoepithelial lesions [35]. Neoplastic cells are heterogeneous, including small centrocyte-like lymphocytes with nuclei that may be regular or incisive and moderately dispersed chromatin and sparse cytoplasm, variable lymphocytes plasmacytoid, and plasma cells. Immunoblasts and centroblasts are frequently present. Lymphoepithelial lesions are common and are characterized by neoplastic cells that infiltrate and destroy the ciliated bronchial and bronchiolar epithelium. Typically, BALT lymphoma shows positive for CD19, CD20, CD22, and CD79a and negativity for CD3, CD5, CD10, CD23, BCL6, and cyclin D1 [3,4,9,29]. CD5 or rarely BCL6 or CD10 can be expressed by cancer cells [1,36]. The Ki-67 proliferation index is low, normally being below 10% and typically it is higher in the residual germinal centers. Frequently, to make a diagnosis, it is necessary to verify the restriction of the kappa and lambda light chains of the plasma cells and/or B-lymphocytes. It is possible to find, in 50% of cases, the presence of the chimeric protein (*BIRC3-MALT1*) and the transcriptional deregulation of *BCL10*, *MALT1*, and *FOXP1* [37]. Although unspecific, detection of chromosome 3 and 18 trisomy is common. The most common cytogenetic abnormality is t(11;18)(q21;q21)(31–53%), resulting in a fusion transcript *API2-MALT1* which determines the activation of the *NF-kB* pathway [38]. This translocation is often detected in the lung and gastric EZML. New genetic alterations, such as chromatin remodeling, *BCR/NF-kB*, and *NOTCH* pathways, along with recurrent *TET2* inactivation, have been described in recent studies through the use of next-generation sequencing analyses [39]. Computed tomography (CT) is the first-choice investigation to evaluate pulmonary EMZL. In most cases, they often present as lung solitary or multiple accidental nodules less than 5 cm in size, mimicking other pathologies such as, for example, pulmonary adenocarcinoma and infections [40]. In the presence of lungs or upper airways primary involvement or *H. pylori* infection, consider an endoscopy to detect silent lymphoma localization. The low specificity of imaging is frequently associated with a delay in diagnosis. In case both lungs are involved, the picture can simulate lymphomatoid granulomatosis. Other radiological findings can be found: infiltrates with bronchograms, consolidation area with apparent cavitations, or a peripheral mass with pleural involvement. In most patients, the lesions remain stable for several years. Patients with pulmonary EMZL may present with hilar and/or mediastinal lymphadenopathy in about 30% of cases. Bronchiectasis is a typical lesion; necrosis is not common, while pleural effusion is present in 10% of patients. FDG-PET has a high sensitivity, with avidity rates between 80–100%; however, its use for staging remains debated [41].

EMZL enters differential diagnosis with benign reactive processes and with other B-cell lymphomas, such as small lymphocyte lymphoma and lymphoplasmacytic lymphoma. The crucial diagnostic procedure is the biopsy examination, even if frequently the material of the needle biopsy is insufficient and with artifacts. Immunocytochemistry and the research for the rearrangement of the MALT1 gene are useful for correct differentiation [35,36].

### 4.3. Ocular Adnexal Marginal Zone Lymphoma (OAML)

Ocular adnexal lymphoma (OAL) accounts for 1–2% of all NHLs and 5–10% of all extranodal NHLs; 80% of OAL belongs to EMZL [42]. Among the risk factors predisposing the onset of ocular EZML, we find the environment, occupational exposure, autoimmune diseases, and infectious agents. It should be noted that the incidence of this lymphoma has increased in the last few decades. The age of onset is approximately 65 years, with a female prevalence. The median between symptom onset and diagnosis is approximately 6–7 months. The clinic is very heterogeneous, being influenced by the site involved: 25% have conjunctival lesions and 75% intraorbital masses; bilateral involvement is reported in 10–15% of cases [43].

There is no evidence of normal lymphatic tissue in the orbital region, and it is uncertain whether MALT is present in the normal conjunctiva. However, the onset of MALT may be caused by the presence of chronic infections and/or autoimmune diseases. OAL has the classic morphological and immunophenotypic characteristics of most EZMLs. A heterogeneous cellular expansion of centrocytes, monocytoid cells, and small lymphocytes is commonly found at microscopic examination. Immunophenotype shows positivity for CD20, CD79a, IgM, PAX5, BCL-2, TCL1, IRTA1, CD11c, CD43, CD21, and CD35 and negativity for IgD, CD3, CD5, CD10, CD23, cyclin D1, BCL-6, and MUM1 [1]. There is a frequent finding of reactive T cells and reactive germinative centers. In 55% of cases, the PCR analysis shows monoclonal immunoglobulin heavy chain and somatic hypermutations in 60%. The most accredited hypothesis is that OAL derives from a clonal expansion of the post-germinal center B cells [44]. Differently from the gastric EZML, the t(11;18)(q21;q21) is detectable only in 3% of cases [45]. FISH analyses showed the presence of aneuploidy, trisomy 3, and/or 18. It is possible to detect the MYD88 L256P mutation [46].

Johansson et al., in a population of 82 OAMLs, performed whole genome sequencing (WGS) and/or whole exome sequencing (WES) for 13 OAML cases and sequenced 38 selected genes. In 11% of cases, the presence of the JAK3 gene mutation was highlighted, resulting in a reduced PFS compared to non-mutated cases. Other mutations found in 5–10% of cases involved members of the collagen family (collagen type XII alpha 1/2 (*COL12A1, COL1A2*)) and *DOCK8*. In WGS were described loss of *TNFAIP3*, recurrent gains of the *HES4 NOTCH* target, and of members of the *CEBP* transcription factor family [47].

Jung et al. analyzed 10 cases of OAML by sequencing the entire genome and RNA and a further 38 cases using targeted sequencing. In this study, the alterations involved genes associated with the activation of the nuclear factor *NF-kB* pathway (60%), chromatin modification and transcriptional regulation (44%), and B lymphocyte differentiation (23%). Furthermore, the sequencing of the whole genome showed the elimination of 6q23.3, a region containing *TNFAIP3* in 50%. In targeted sequencing, a *TNFAIP3* mutation was the most common alteration (54%), followed by mutations in *TBL1XR1* (18%), *cAMP* response element binding protein (*CREBBP*) (17%), and *KMT2D* (6%). *TBL1XR1* mutations were located within the WD40 domain and resulted in increased binding of *TBL1XR1* to the nuclear receptor corepressor (NCoR), leading to increased NCoR degradation and activation of NF-kB and JUN signaling pathways [48].

In another study, Johansson, in a population of 63 OAMLs, showed mutations of *NOTCH1* and *NOTCH2* in 8% of cases, indicating their role in the pathogenesis and in 22% the mutation of *KMT2D*. Furthermore, *MYD88* mutations have been associated with inferior DFS [49].

Generally, the conjunctival localizations of EZML show the characteristic “salmon-red lesion”, while the intraorbital sites can manifest themselves with exophthalmos (27% of cases), palpable masses (19%), ptosis (6%), diplopia (2%), edema, or orbital nodules [43]. Impaired ocular motility can occur. A correct and experienced histological diagnosis is crucial.

For a correct staging of the disease, it is advisable to perform CT of the neck, thorax, and abdomen-pelvis, which may be sufficient for the evaluation of any systemic involvement. PET/Tc examination has low sensitivity. In patients without cytopenia, bone marrow biopsy can be omitted. Magnetic resonance imaging (MRI) is the ideal method for evaluating the orbital region. Unlike the other lymphomas, which follow the Ann Arbor staging, the TNM classification is regularly used in ocular EZML [50].

The association between autoimmune diseases and OAL has been evaluated in several studies, showing a higher incidence, especially in women, of thyroid disease. Autoimmune diseases have been found mostly in non-gastric EZMLs, such as salivary glands and ocular adnexa; however, their presence does not affect the clinical course [51,52,53].

A pathogenic role of *C. psittaci* has been suggested in several studies. *C. psittaci* persistent infection induces proliferation of polyclonal cells and causes resistance to apoptosis of infected cells [54,55]. In addition, *C. psittaci* DNA was found in 80% of biopsy sections of OAML patients [56]. The presence of *C. psittaci* was also found in vitro cultures from conjunctival swabs and peripheral blood [57]. This association is also supported by epidemiological data: in fact, patients with OAML reside in rural areas where a history of chronic conjunctivitis and the presence of domestic animals are frequently reported [57]. Another evidence of this association is the fact that OAML tends to regress after doxycycline therapy in *C. psittaci*-positive patients [58]. No further data are showing the association of OAML with other infections.

### 4.4. Other EMZL Sites

Other localizations of MALT lymphoma are rare in comparison to the ones discussed above, given the diffusion of lymphoid tissues in the whole organism; potentially any tissue could become a site involved [1].

The salivary gland is the most frequent among rare sites, followed by skin (10%), thyroid (5%), liver (3%), breast (3%), genitourinary tract, thymus, dura mater, and other rare sites [2,3].

Despite a relationship between infections and autoimmune disorders being well known, if eradication of pathogens or management of the autoimmune disease is not pursuable, a different treatment approach must be attempted in symptomatic patients.

The pathogenesis shares the same model of that of the more diffuse variants; the genetic mutations found are similar and so is the relation with infections and autoimmune disorders. The linkage between chronic infection/inflammation and etiopathogenesis has been demonstrated in these entities as well. Sjogren syndrome has been associated with salivary glands MALT, pathogens such as HCV (liver), *Borrelia Burgdoferi* (skin), and *Campylobacter jejuni* (small intestine) Figure 1 and Table 1 [4,5].

The clinical presentation follows an indolent course but can vary based on sites involved and lesions size. Despite the site involved, the disease can be diffused at diagnosis with variable FDG uptake at PET-CT, which can detect the disease in around 71% of patients [59].

Staging could be adapted depending on the organ involved, to perform a more specific investigation. The diagnostic workup for salivary glands EMZL should include Anti-SSA/Ro and anti-SSB/La antibodies, Ear, Nose, Throat, ENT examination, echography, and endoscopic procedure, and Mammography or Breast MRI should be performed in Breast Malt [27].

## 5. Treatment of EMZL

In all primary EMLZ sites, consider the initial staging with clinical evaluation, hemogram and cytological blood analysis; biochemical, renal, and liver tests, with lactate dehydrogenase; serum (urine) protein electrophoresis; HBV, HIV, and HCV serologic tests; possibly completed by a PET-CT to assist in medical decision-making, in biopsy site, staging, fear of histological transformation; and bone marrow aspirate and biopsy in case of doubt about bone marrow involvement (cytopenia, bone hyper fixation in PET). The choice of treatment is based on various factors related to the patient: age, comorbidities, performance status, and life expectancy; to the disease: tumor size, location, related symptoms, and ocular impairment; and to the treatment itself: objective, side effects, sequelae, and drug availability. In consideration of these variables, the clinician will choose between (i) chemotherapy, (ii) surgery, (iii) immunotherapy, (iv) radiotherapy (RT), or (v) watch and wait for (W & W) [19,27]. Surgery is essential for diagnosing EMZL and is considered therapeutic for patients with localized disease. In isolated cases of OAML, it can be considered as an initial therapeutic approach. Complete removal of the lesion is often fraught with complications and is therefore often not recommended.

W & W approach can be adopted for asymptomatic patients not fulfilling common criteria for treatment. It can be used in patients with modest residual disease after surgery, RT, and oron antibiotic therapy, nevertheless considering the question of adjuvant treatment [27]. For symptomatic patients or critical localization of disease, treatment must be started. Radiotherapy (RT) is a well-consolidated strategy for localized disease in previously untreated patients, diversifying the modality according to the site of localization. RT allows optimal disease control (2-year PFS 100%, 4-year PFS 89%) in EMZL stages I–II. The dose recommended is 24 Gy, but a low-dose schedule (4 Gy) may also be considered in the frail/palliative setting or for the involvement of critical sites [27,60,61]. Antibiotic/target therapy is essential in the management of EMZL related to specific pathogens such as *C. psittaci*, *H. pylori*, *C. jejuni*, and HCV. A standard immunochemotherapy treatment is generally used for symptomatic patients affected by advanced EMZL of various origins (BALT, advanced MALT, etc.) [5,21,27]. IELSG-19 trial tested the combination Rituximab and Chlorambucil in 401 patients, with a 5-year PFS of 72% and 5-year OS of 90%. The BELTAMO study reported the experience of 60 patients treated with R-Bendamustine (7-year PFS: 92.8%, 7-year PFS: 92.8%) [61,62,63]. R-CHOP/CVP can also be considered [64]. Lossos et al. [65] also have tested Radioimmunotherapy in this setting. Immunomodulating agents, such as thalidomide and lenalidomide, have been tested in EMZLs and can be considered for further lines [66]. The rituximab–Lenalidomide combination was tested in a cohort of patients with advanced EZML and showed an ORR of 89% and a CR of 67% in previously untreated patients [67]. Recently, Bruton-Kinase inhibitors (BTK), such as Ibrutinib, have been tested on MZL, showing efficacy on previously treated patients (*n* = 30), duration of response, and PFS at 33 months was 50% and 26%, respectively [68].

## 6. Treatment of Particular Entity

### 6.1. Gastric MALT Lymphoma

After proper staging, performed as mentioned in the previous paragraph, the distinction between *H. pylori*+ and *H. pylori*− patients must be performed [50,69]. The relationship with *H. pylori* brought bacteria eradication as the gold standard for both diseases. Immunotherapy employment is currently under investigation to be the backbone therapy of relapsed/refractory patients [27,70,71,72]. Management of MALT patients is summarized in Figure 2.

Antibiotic therapy should be targeted on the regional resistance spectrum of the bacteria. With proper treatment, a complete remission rate ranging from 75 to 80% is expected within 24 months [73]. Usually, regimens combine proton pump inhibitors plus two antibiotics. Response rate should be irrespective of stage, and no further treatment is required. For *H. pylori*-negative patients given the inferior outcome of *H. pylori* eradication, the immediate use of anti-lymphoma treatment can be considered. In recent years, the incidence of *H. pylori* “negative” gastric MALT has risen from 5 to 10% to 30–50%. Patients must be considered negative with caution, given the possible adaptive mechanisms of the bacteria to the tumor environment, such as a change to coccoid forms and migration to other sites. Negativity to *H. pylori* is considered as the absence of the bacteria on histology, together with negative breath test/stool antigen and serologic absence in the bloodstream [72,73]. In this subset of patients, other infections sustaining the disease, such as Helicobacter subspecies, could be detectable and are likely to respond to antibiotic treatment, as well as false-negative tests. Autoimmune disease tests must be taken into account. For these patients, the effectiveness of antibiotic treatment is considerably lower. Patients achieving a complete response do not require further treatment. In addition, in case of distant relapse or persistent *H. pylori*, a second anti-*H*. *pylori* regimen should be considered. Clarithromycin 500 mg × 2 daily can be considered as a direct antineoplastic treatment; it has shown higher efficacy in gastric MALT (55% ORR, 24% CR) than other subtypes [27,74].

In localized MALT patients, antibiotic treatment must be preferred; the histological response can be slow, and proper timing for histological assessment must be considered. For refractory patients, an involved site (IS) RT proved to achieve 100% CR in post-RT endoscopic biopsies with 100% OS and Disease-Free Survival at 2 years; a dose superior to 24 Gy was not related to better outcomes [75].

Furthermore, low-dose radiotherapy (LDRT) with 4 Gy (2 Gy × 2) was tested in MALT and localized MZL with favorable outcome (2 years PFS 85% and OS 91%) [76]. In the presence of t(11;18)(q21;q21), the outcome expected with antibiotics is poorer, and radiotherapy can be secondarily associated.

In MALT patients, lenalidomide, in monotherapy or combination with rituximab, showed promising results and can be considered a valid option for refractory/relapsed patients. A delayed response with immunomodulatory therapy is not rare [66,67,77]. The inclusion of other MALT subtypes or indolent lymphomas must be considered as a possible bias of recent protocols. BTK-inhibitor ibrutinib, which showed an ORR of 53% and 62% PFS at 18 months, can be considered even if employed in an extensive marginal zone trial [68]. Chemotherapy showed high activity against MALT-lymphoma with the employment of various agents. While rituximab monotherapy could be avoided to minimize the risk of emergency of CD20-negative clones. The combination of rituximab plus oral chlorambucil showed better event-free survival (68% at 5 years of follow-up vs. 50% for rituximab alone) [66,67].

Treatment response evaluation is still based on histological assessment of new biopsies. An early breath-test, in those cases *H. Pylori* related, can be performed after 4 weeks of antibiotic therapy. Mucosal monoclonal B lymphocytes may persist even with histological regression. Endoscopic follow-up is mandatory. First biopsies must be performed within 2–3 months from the end of treatment and, afterward, twice a year for the first 2 years. Lymphoid infiltration complicates histologic evaluation, and transient local relapses can occur; nevertheless, in absence of reinfection, they are self-limiting. Endoscopic and systemic follow-up is recommended once a year from the third year, considering, moreover, that the risk of gastric adenocarcinoma is sixfold higher than the general population [27,72].

### 6.2. BALT Lymphoma

BALT has an indolent course with 5 and 10-year survival rates of 90% and 70%, respectively, and a median survival greater than 10 years [78]. As mentioned above, proper staging is mandatory with bronchoscopy, bronchoalveolar lavage, and possibly PET/CT. There are several possible treatments: surgical resection for localized lesions, watch and wait, and radiotherapy or chemotherapy in unresectable cases. Management of BALT is highlighted in Figure 3. In the case of localized disease, a local approach with surgery or radiotherapy is preferable. In this background, these procedures exhibit significantly better PFS than those receiving systemic treatment.

Regarding BALT, Rituximab monotherapy is associated with a good response, approximately 70%, but has a high risk of relapse, about 30% [79]. Chemo-immunotherapy remains the backbone for advanced patients as reported above [32,66,67,68]. Studies are currently underway to evaluate the efficacy of new therapeutic strategies. For example, Noy et al. recently published data on the safety and efficacy of Ibrutinib in a cohort of patients who relapsed or are refractory to rituximab [66]. Preliminary data showed excellent activity of the PI3K inhibitors copanlisib and umbrasilib [80].

At the end of treatment, the patient must undergo a total body CT scan and, if appropriate, PET examination. Thereafter, the checks must be performed every six months for the first five years and once a year thereafter. The choice of treatment for disease recurrence is based on the type of previous therapy, time to relapse from previous treatment, disease status, and performance status [27].

### 6.3. Treatment: OAML

OAML has an indolent clinical course; in most cases, patients have slow progression with no signs of infiltration and impairment of ocular function. It has an excellent prognosis with an overall 5-year survival > 90%, although recurrence is quite frequent. The 5-year PFS is 65%. Due to the rarity of the disease, there are no standardized guidelines [80,81].

Moreover, in this setting, a therapeutic approach could include the strategies mentioned above; treatment approach is summarized in Figure 4. Surgery is essential for diagnosing OAL. In isolated cases, it can be considered as an initial therapeutic approach. Complete removal of the lesion is often fraught with complications, and therefore, it is not often recommended [82].

Radical surgery can be considered in case of localized sites such as conjunctiva or lacrimal glands. In case of incomplete excision, an adjuvant treatment should be taken into account. In retrobulbar localization of the lacrimal or conjunctival gland, it is advisable to irradiate the entire orbit because it is associated with a lower recurrence rate than partial irradiation [83]. Some studies have shown a reduction in locoregional relapses, with a 5-year PFS equal to 60–65% years with a radiotherapy approach from 24 to 25 Gy. However, contralateral relapses have been observed in up to 40% of cases [84]. Early toxicity is generally manageable, while late toxicity, such as cataracts and xerophthalmia, affects 50% of patients. 36 Gy doses have been observed to be associated with more severe complications such as ischemic retinopathy, neovascular glaucoma, and vision loss [85]. To reduce the toxic effects of radiotherapy at the ocular level, it is possible to evaluate the use of radiotherapy with 4 Gy. Fasola et al. reported a case series of 27 patients with NHL of the ocular adnexa treated with a total radiation dose equal to 4 Gy, obtaining a 2-year local control rate equal to 100% [86].

Ocular EZML exhibits CD20 expression and therefore may benefit from therapy with rituximab alone or in combination with chemotherapy regimens. In OAML, in consideration of the association between ocular EZML and *C. psittaci* infection, antibiotic therapy is aimed at eradicating the bacterial infection that determines the chronic antigenic stimulation related to the onset of lymphoma. A prospective study of 27 patients treated with doxycycline showed an overall response rate (ORR) of 48:64% for *C. psittaci* DNA positive patients and 38% for negative patients [87]. The 2-year PFS was 66%. A further Korean study showed a 5-year PFS of 65% in patients treated with antibiotic therapy. These data demonstrated to us how doxycycline is an effective, safe, and mainly active therapy in patients with *C. psittaci* DNA positive [88]. Several studies have shown the efficacy of Rituximab alone, especially in conjunctival locations [89]. The association with radiotherapy did not result in significant benefits [90]. Given the low toxicity, rituximab monotherapy should be considered in patients unsuitable for more intensive treatments. Data have recently been published showing the efficacy of intralesional rituximab [91]. Some studies have shown interesting data with anti-CD20 radioimmunotherapy [65].

In consideration of the rarity of the disease, the studies conducted on OAMLs relating to chemotherapy or immunochemotherapy are small and retrospective, and therefore, they should be considered in selected cases [65,66,67]. Both anthracycline-containing regimens and purine analogs alone or in combination are burdened by significant hematological toxicity, limiting their use in daily clinics [64].

A retrospective study on 689 OAML patients was recently published: in the early stages IE external beam radiation therapy (EBRT) monotherapy showed a better disease-specific survival (DSS) at 10 years compared to chemotherapy, while in stages IIIE/IVE, DSS was higher in patients treated with rituximab-chemotherapy regimens than in chemotherapy alone [92].

Three months after the end of treatment, the patient must perform an ophthalmologist examination, magnetic resonance, and, if appropriate, PET/CT examination. Subsequently, the checks must be set every six months for the first five years and once a year thereafter [27].

### 6.4. Other EMZL

Response rates expected are high, and the data coming from the NF10 FIL study highlight a rate of Progression of Disease within 2 years (POD24) of 16% and 5-Year PFS for EMZL [93].

Localized disease should be approached like other sites of origin, including evaluation of W & W, surgery, and radiotherapy. For advanced disease, chemoimmunotherapy is considered a valuable approach [27,80].

Chlorambucil showed excellent ORR in EZML, both alone and in combination [26,64]. The IELSG19 trial, which involved MALT lymphoma patients who failed or were not eligible for local therapy, compared chlorambucil vs rituximab vs rituximab in combination with chlorambucil. The combination therapy arm had better data in complete response rate and 5-year EFS [66].

The rituximab-bendamustine combination was evaluated in a phase II GELTAMO study and had an ORR of 100% and 4-year EFS of 88% [67].

Both anthracycline-containing regimens and purine analogs alone or in combination are burdened by significant hematological toxicity, limiting their use in daily clinics [66].

Patients with EMZL at non-gastric sites at the end of treatment must undergo specific assessment depending on the origin site and a complete evaluation for indolent lymphomas. Clinical, laboratory, and imaging assessment should be performed every 3 months for the first two years and twice a year thereafter for 10 years. Biopsy of residual lesions is recommended [27].

For the treatment of relapsed or refractory patients, there is no consensus. Where possible, it is recommended to include the patient in clinical trials. If this is not possible, treatment should be based on several factors including the type of previous therapy, time to relapse from previous treatment, disease status, and performance status.

## 7. Conclusions

In this review, we have tried to describe the most common forms of EZML, which account for 70% of MZL. EZML is a heterogeneous disease with a variable clinical presentation. Although much progress has been made in the biological understanding of marginal zone lymphomas with the identification of molecular and cytogenetic mechanisms, these currently have no impact on the therapeutic approach. There are several lines of treatment. However, except for *H. pylori* eradication therapy in localized gastric lymphoma, selecting an appropriate treatment can be difficult and requires the development of new studies, which could allow target employment of new agents.

## Figures and Tables

**Figure 1 cancers-14-01742-f001:**
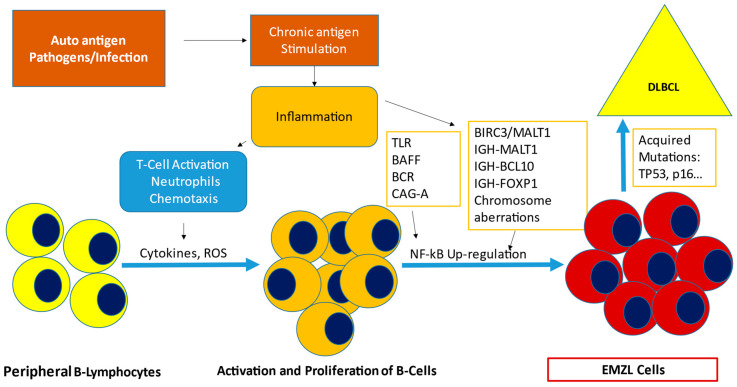
Etiopathogenesis of Extranodal Marginal Zone Lymphoma. Chronic antigen stimulation is given by associated infections (*H. pylori*, *C. Psittaci*, etc.) and/or autoimmune diseases and sustains inflammation. T-Cell activation and Neutrophils chemotaxis are enhanced, producing cytokines and Reactive oxygen species (ROS), which are responsible for peripheral B-Lymphocytes activation and proliferation. Inflammation causes the upregulation of NF-kB pathway, through Toll-like receptors (TLR), B cell receptor (BCR), B-cell activating factor (BAFF), and the exposure of DNA to the damaging effects of ROS. Mutations and Genomic aberrations can occur in this context, promoting tumorigenesis and transformation into Extranodal Marginal Zone Lymphoma (EMZL) cells. Acquired mutations can occur, such as deletions of p53 and or p16, leading to DLBCL transformation. CAG-A, Cytotoxin-associated antigen A; BIRC3/MALT1, t(11;18); IGH-MALT1, t(14;18); IGH-BCL10, t(1;14); IGH-FOXP1, t(3;14); DLBCL, Diffuse Large B-Cell Mutations.

**Figure 2 cancers-14-01742-f002:**
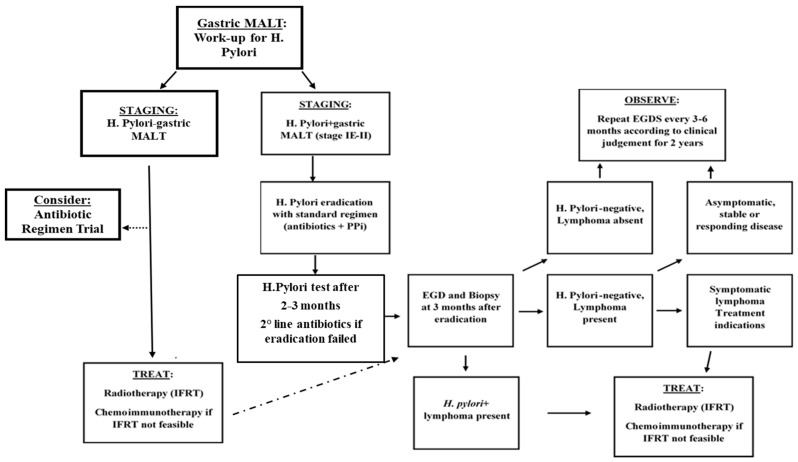
Treatment algorithm for stage IE-II gastric-MALT Lymphoma. *H. pylori*, *Helicobacter pylori*; PPi, Proton pump inhibitors; EGDS, Esophagogastroduodenoscopy; IFRT, Involved field radiotherapy.

**Figure 3 cancers-14-01742-f003:**
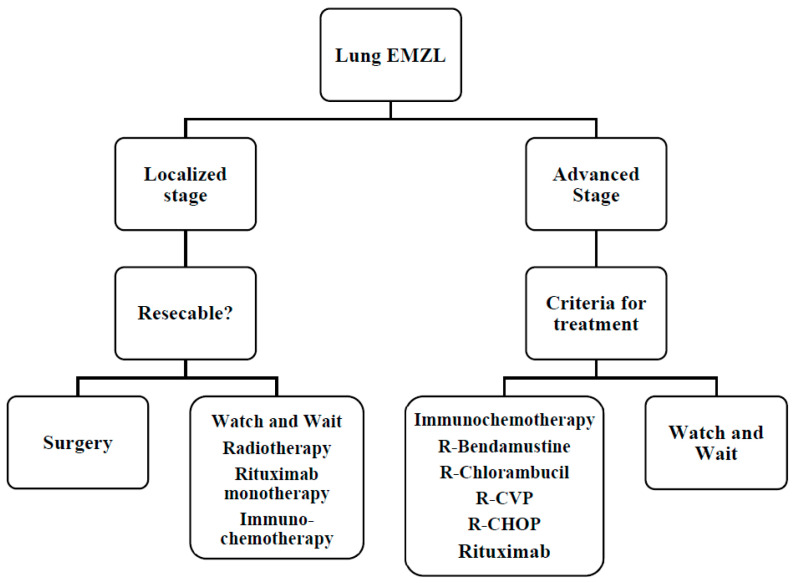
Therapeutic algorithm for lung EZML. R-CHOP, Rituximab, Cyclophosphamide, Vincristine, Adriblastine, Prednisone; R-CVP, Rituximab, Cyclophosphamide, Vincristine, Prednisone.

**Figure 4 cancers-14-01742-f004:**
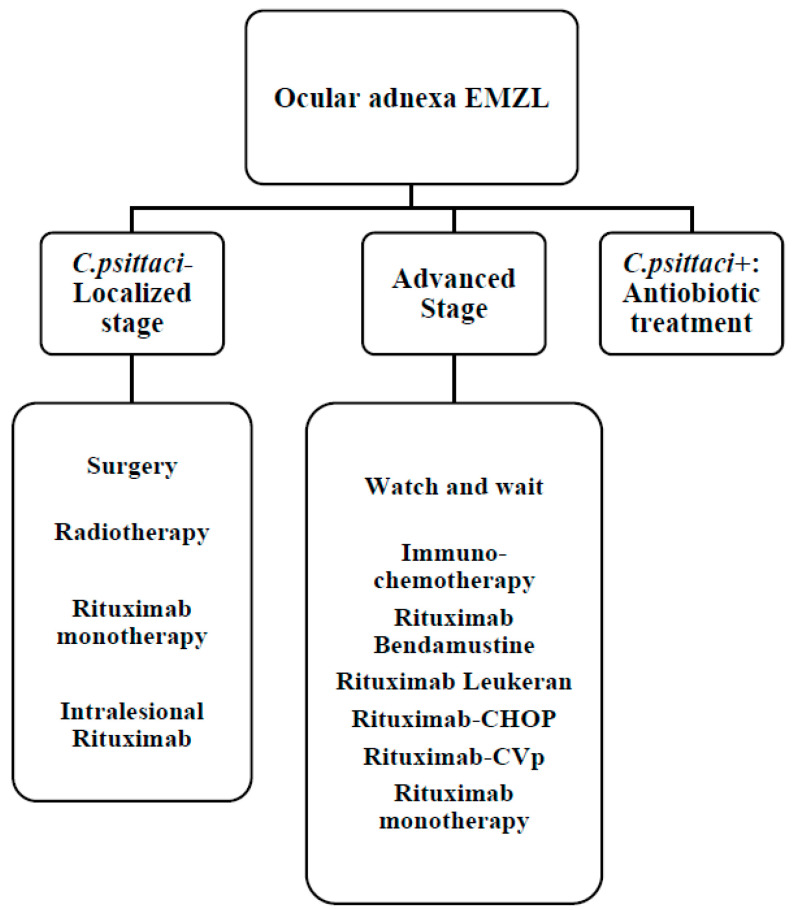
Therapeutic algorithm for ocular adnexa EZML. *C. psittaci*, *Chlamidophila psittaci*; R-CHOP, Rituximab, Cyclophosphamide, Vincristine, Adriblastine, Prednisone; R-CVP, Rituximab, Cyclophosphamide, Vincristine, Prednisone.

**Table 1 cancers-14-01742-t001:** Site-related pathogens and genetic mutations.

EMZL Site-Related Genomic Aberrations, Pathogens, Autoimmune Disorders		
Localization	Genomic Aberrations	Pathogens/Autoimmune Disorders
Gastric MALT	Trisomy: +3, +18	*Helicobacter pylori* *Helicobacter Heimanni*
	IGH/MALT1 t(14;18)(q32;q21) BIRC3/MALT1 t(11;18)(q21;q21) BCL10/IGH t(1;14)(p22;q32)	
Intestine	Trisomy: +3, +18	*Campylobacter jejuni*
	BIRC3/MALT1 t(11;18)(q21;q21) BCL10/IGH t(1;14)(p22;q32)	
Skin	Trisomy: +3, +18	*Borrelia burgdorferi*
	IGH/MALT1 t(14;18)(q32;q21)	
	FOXP1/IGH t(3;14)(p14.1;q32)	
Ocular Adnexa	Trisomy: +3, +18	*Chlamydia psittaci* *Sjogren’s syndrome*
	IGH/MALT1 t(14;18)(q32;q21)	
	FOXP1/IGH t(3;14)(p14.1;q32)	
	TNFAIP3 target mutation	
Lung	Trisomy: +3, +18	*Achromobacter xylosoxidans* *Lymphocytic interstitial pneumonia*
	BIRC3/MALT1 t(11;18)(q21;q21)	
Salivary Gland	Trisomy: +3, +18 IGH/MALT1 t(14;18)(q32;q21)	*Sjogren’s syndrome* *Hepatitis C virus*
	TBL1XR1 target mutation	
	GPR34 target mutation	

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
