# Peer review of "Extranodal Marginal Zone Lymphoma: Pathogenesis, Diagnosis and Treatment"

_cancers, 2022, doi:10.3390/cancers14071742_

Round 1

Reviewer 1 Report

In this review paper authors attempted to summarize the current knowledge about extranodal marginal zone lymphomas (EMZL). The manuscript quality is very heterogeneous. The first three pages are rather poorly written with quite some factual, logical and grammatical errors, whereas starting from the paragraph 4.1 the quality and presentation of information increases considerably. I suggest that the authors could revise the entire structure of the manuscript remove some low quality parts, which are sometimes duplicating information presented later. This would make the paper more even and better understandable for the reader.

While clinical aspects and treatment options are discussed rather well, the manuscript lacks more detailed and more up to date description of the genetic background of the disease including the recently published systematic studies on this topic. Also, a tabular representation of genetic data e.g. with genetic aberration / EMZL subtype predominance (if present) and frequency of occurrence would be desirable.

Below some of isolated deficiencies are highlighted, but the list is non-exhaustive as they were numerous. Systemic revision of the entire manuscript is of paramount importance. Also, there are quite some grammatical mistakes left in the text especially first pages, so it would definitely profit form a professional language editing.

The first sentence of the abstract needs to be corrected: „<…> median age of 60 years <…>“ is meaningless. Probably the median age at diagnosis is meant and the sentence have to be supplemented accordingly. Moreover, the term „non-Hodgkin lymphoma“ is no longer an official definition according to WHO classification, therefore I recommend avoid using it here and further in text.

In paragraph Epidemiology mean age greater than 60 years is mentioned. Most probably the median and not the mean was meant?

Abbreviations for Helycobacter pylori or Chlamydophila psittaci like „Hp“ and „Cp“, respectively are rather unusual and not desirable. I would advise to abbreviate only the genus and write out the species fully, e.g. H. pylori. here and elsewhere in the text.

In the sentence «Once the underlying infection or autoimmune disorder is detected, specific treatment is as well responsible for tumor remission» I would replace the word «responsible» to «leads to» or «contributes to».

The sentence „On genetic analysis, translocations of trisomy 3, isochromosome 17q and 2p11 may be present [1] is wrong and confusing in multiple places. First, what is translocations of trisomy 3? Also it is not clear what is meant with 2p11. Is it a deletion, duplication or some other aberration of this genetic region?

For me it is not clear, what the following sentence means: „It has nowadays commonly consolidated the model in which chronic antigenic exposure and inflammation sustain the development of the disease is consolidated“

In the sentence „MALT cells are similar to plasma cells (CD20-) and often harbor the same translocation t(11;18)(q21,q21) [24]“ what does the word „same“ refer to? Besides, according to the ISCN nomenclature the translocations should be written without spaces, and chromosomes and chromosomal bands are separated by semicolon, not comma. Please correct in the entire manuscript.

The aesthetic quality of figures 2, 3 and 4 is low and needs to be improved considerably

Author Response

Dear Revisor 1,

Thank you for your comments, we tried to address all your suggestions, surely improving of the work. Find the revised paper attached

Here you will find point by point the changes. All the modifications in the text addressed to your specific observations are highlighted in light blue

In this review paper authors attempted to summarize the current knowledge about extranodal marginal zone lymphomas (EMZL). The manuscript quality is very heterogeneous. The first three pages are rather poorly written with quite some factual, logical and grammatical errors, whereas starting from the paragraph 4.1 the quality and presentation of information increases considerably. I suggest that the authors could revise the entire structure of the manuscript remove some low quality parts, which are sometimes duplicating information presented later. This would make the paper more even and better understandable for the reader. While clinical aspects and treatment options are discussed rather well, the manuscript lacks more detailed and more up to date description of the genetic background of the disease including the recently published systematic studies on this topic. Also, a tabular representation of genetic data e.g. with genetic aberration / EMZL subtype predominance (if present) and frequency of occurrence would be desirable.

Below some of isolated deficiencies are highlighted, but the list is non-exhaustive as they were numerous. Systemic revision of the entire manuscript is of paramount importance. Also, there are quite some grammatical mistakes left in the text especially first pages, so it would definitely profit form a professional language editing

Thank you for your contributions, it helped to improve the work. The paper was revised by a mother tongue and grammar and logical errors were corrected along the paper. We improved the genetic background section (paragraph 2.1), and revised the whole paper to make it more understandable and more communicative.

The first sentence of the abstract needs to be corrected: „<…> median age of 60 years <…>“ is meaningless. Probably the median age at diagnosis is meant and the sentence have to be supplemented accordingly. Moreover, the term „non-Hodgkin lymphoma“ is no longer an official definition according to WHO classification, therefore I recommend avoid using it here and further in text.

The first sentence was corrected and nomenclature revised.

In paragraph Epidemiology mean age greater than 60 years is mentioned. Most probably the median and not the mean was meant?

The text was corrected

Abbreviations for Helycobacter pylori or Chlamydophila psittaci like „Hp“ and „Cp“, respectively are rather unusual and not desirable. I would advise to abbreviate only the genus and write out the species fully, e.g. H. pylori. here and elsewhere in the text.

Abbreviations were changed.

In the sentence «Once the underlying infection or autoimmune disorder is detected, specific treatment is as well responsible for tumor remission» I would replace the word «responsible» to «leads to» or «contributes to».

Sentence was modified to improve the communication

The sentence „On genetic analysis, translocations of trisomy 3, isochromosome 17q and 2p11 may be present [1] is wrong and confusing in multiple places. First, what is translocations of trisomy 3? Also it is not clear what is meant with 2p11. Is it a deletion, duplication or some other aberration of this genetic region?

Phrase and sentence were modified.

For me it is not clear, what the following sentence means: „It has nowadays commonly consolidated the model in which chronic antigenic exposure and inflammation sustain the development of the disease is consolidated“

The sentence was modified.

In the sentence „MALT cells are similar to plasma cells (CD20-) and often harbor the same translocation t(11;18)(q21,q21) [24]“ what does the word „same“ refer to? Besides, according to the ISCN nomenclature the translocations should be written without spaces, and chromosomes and chromosomal bands are separated by semicolon, not comma. Please correct in the entire manuscript.

Sentence was revised and nomenclature corrected

The aesthetic quality of figures 2, 3 and 4 is low and needs to be improved considerably

Quality of figures was improved.

Reviewer 2 Report

Manuscript ID: cancers-1602859
Type of manuscript: Review
Title: Extranodal Marginal Zone Lymphoma: Pathogenesis, Diagnosis and
Treatment
Authors: Alice Di Rocco, Luigi Petrucci, Giovanni Manfredi Assanto, Maurizio
Martelli, Alessandro Pulsoni
The article by Alice Di Rocco et al. is a good review article focused on the management of marginal zone lymphoma with initial mucosal localization.
This review has epidemiological significance because marginal zone lymphoma is the third most common non-Hodgkin lymphoma, after diffuse large cell B-cell lymphoma and follicular lymphoma. It accounts for 50-70% of marginal zone lymphomas and 7-8% of non-Hodgkin's lymphomas.
The article considers its clinical and biological, cytogenetic and genetic aspects, the extension assessment and the therapeutic management adapted to the site affected by the MALT lymphoma.

COMMENTS:
2. Epidemiology

In order to group the message:
MZL accounts for approximately 10% of non-Hodgkin's lymphomas with a mean age greater than 60 years at diagnosis. In the United States, the incidence is 18.3 cases per 1 million person-years [2].

Add : EMZL accounts for 50-70% of marginal zone lymphomas and 7-8% of non-Hodgkin's lymphomas. The most affected sites are the stomach, ocular adnexa, lung and salivary glands.

2.1 Etiopathogenesis

1- Emphasize the importance of the presence of complete or partial trisomies of the long arm of chromosomes 3 and 18 (+3/+3q or +18/+18q) which are common to all subgroups of LZM but their association is important for differentiate LZMs from other SLP-Bs.

2- Cytogenetic abnormalities could be brought together in a specific paragraph. A trisomy of chromosome 3 or 8 is highly suggestive of marginal zone lymphoma but is not specific. Other non-specific anomalies to be specified in this paragraph such as gain of 6p25 and ocular adnexa EZML….

3- Specify right from the start in this paragraph that the incidence of translocations varies according to the anatomical site: t(1;14)(p22;q32) inducing BCL10-IGH is found more in the pulmonary and intestinal MALTs; the t(14;18)(q32;q21) inducing IGH-MALT1 is found more in ocular lymphomas then of the salivary glands, the t(3;14)(p14;q32) inducing FOXP1-IGH is found mainly in those of the thyroid then in the MALT of the orbit and the skin.

4- Express the molecular fusion in agreement with the carrier chromosome.

5- « Once the underlying infection or autoimmune disorder is detected, specific treatment is as well responsible for tumor remission ». Consider : …., although antibiotic therapy may have limits in its anti-lymphoma potential in extragastric mucosal sites. Zucca E, Arcaini L, Buske C, et al. Marginal zone lymphomas: ESMO clinical practice guidelines for diagnosis, treatment and follow-up. Ann Oncol 2020;31:17-29. Marginal-zone lymphomas. Rossi D, Bertoni F, Zucca E. N Engl J Med. 2022.

3. Clinical presentation and disease assessment

1- Lung EMZL: Recurrent respiratory infections and/or discovery of lung nodules on imaging studies.

2- Ocular EMZL : slow growing masses, eye redness.

4. Pronostic factors and disease features

1- The MALT International Pronostic Index (IPI)

4.1. EMZL of gastrointestinal tract or gastric MALT

A- Consider the following remarks:
1- Decrease in the incidence of gastric MALT lymphoma with Helicobacter pylori eradication.
2- « It has nowadays commonly consolidated the model in which chronic antigenic exposure and inflammation sustain the development of the disease ». Delete : … is consolidated
3- …. T-regulatoty cells and cytokines … are involved in the inflammation creating the lymphoma pathogenesis together with…
4- « Sometimes immunoglobulins are detectable in false positive immunofixation [23] ». : not clear.
5- ….a high percentage of blasts : replace with « centroblasts ».
6- « Anemia caused by chronic gastric bleeding » : replace with « Iron deficiency anemia ».
7- Mucosal biopsy performed during endoscopic procedure is the gold standard for diagnosis.

- Consider lymphoepithelial lesions in the anatomo-pathology analysis. In mucosal tissues, the clonal marginal zone B cells typically infiltrate the epithelium, forming lymphoepithelial lesions which are considered as a hallmark of MALT lymphoma, although not indispensable for diagnosis.
- Consider the demonstration of monoclonality in order to rule out the diagnosis of inflammation and to confirm lymphoma.

8- Consider endoscopy ultrasonography if possible, in gastric wall infiltration assessment and/or perigastric lymph nodes.
B- « MALT cells are similar to plasma cells » : not clear
C- « Around 25 to 30% of gastric MALT are estimated harboring the BIRC3-MALT1 translocation which guarantees a survival advantage to tumor cells and is absent in pulmonary or ocular adnexa MALTs ».
Yet t(11;18) (q21;q21) / BIRC3-MALT1 can be found in about 40% of pulmonary forms.

In accordance with the statement in paragraph 4.2: “This translocation is often detected in the lung and gastric EZML »

4.2 Bronchus-associated lymphoid tissue (BALT) lymphoma

1- In staging patients with primary involvement of the lungs, or upper airwairs, or Hp infection, consider an endoscopy of the upper digestive tract for lymphoma silent localization.

4.3. Ocular adrexal marginal zone lymphoma (OAML)
Potentially impaired ocular motility. Is not a sentence.

4.4. Other EMZL sites

ENT examination and echography and EGD. Specify the meaning of the abbreviations. ENT: ear, nose and throat ? EGD: oesophagastroduodenoscopy ?

5. Treatment of EMZL

1- In all primary EMLZ sites, consider the initial staging with clinical evaluation, hemogram and cytological blood analysis ; biochemical, renal and liver tests, with lactate deshydrogenase ; serum (urine) protein electrophoresis ; HBV, HIV and HCV serologic tests ; possibly completed by a PET-CT for the purpose of assisting in medical decision-making, in biopsy site, staging, fear of histological transformation; bone marrow aspirate and biopsy in case of doubt about bone marrow involvement (cytopenia, bone hyperfixation in PET).

2- « It can be used in patients with modest residual disease after surgery, RT or antibiothic therapy », add : nevertheless considering the question of adjuvant treatment.

1- To specific Noxa such as.. Noxa meaning ?

2- Recently, Bruton-Kinase inhibitors (BTK), such as Ibrutinib, have been tested on MZL, showing efficacy on previously treated patients (n 30): Duration of response and PFS at 33 months were 50% and 26% respectively [66] Correct puntuation to ensure.

6.1. Gastric MALT lymphoma

A- After “proper staging”: to specify.
Consider the initial staging with esophagogastroduenoscopy; endoscopic ultrasonography il possible; immunohistochemistry for Hp detection; FISH ou PCR for t(11;18) if possible and possibly completed by a PET-CT.

B- Regarding anti-helicobacter pylori antibiotic therapy, grade the message according to validated clinical recommendations and precautionary recommendations.

Also, consider the following data:
1- If the immunohistochemical study for the search for Helicobacter pylori is negative, carry out a fecal antigen or a breath test and a serological study.
2- Prefer anti-infective treatment to gastrectomy with potential sequelae, in Hp+ gastric lymphoma.
3- A second line anti-Hp treatment may be necessary in less than a quarter of cases.
4- In localized Hp+ gastric lymphoma, only an eradicating anti-infectious treatment is put in place, with a study of the histological response which can be slow.
5- Mucosal monoclonal B lymphocytes may persist despite histological regression.
6- Gastric lymphomas with chromosomal translocation t(11;18)(q21;q21) (detection by in situ hybridization FISH or PCR) have a tendency to tumoral progression and show a poor response to antibiotic therapy, even in the presence of the infectious agent. In this case, consider anti-infectious treatment secondarily associated with localized radiotherapy.
7- The anti-infectious treatment will probably have little or no effect in the event of negative detection of Helicobacter pylori associated with the MALT lymphoma.
8- Localized radiotherapy should be considered in the context of localized gastric MALT lymphoma that does not respond to antibiotic therapy or negative for Hp.

6.2. BALT lymphoma

Consider the BALT lymphoma initial staging with bronchoscopy and bronchoalveolar lavage (BAL); possibly completed by a PET-CT and esophagogastroduenoscopy.

1- Figure 3 : The word immunotherapy is regularly cut.
2- Advanced patietns : change to « Advanced patients ».

6.3. Treatment : OAML

1- Consider the OAML initial staging with ophtalmologic clinical evaluation, orbit and head and neck magneric resonance imaging (MRI) ; possibly completed by a PET-CT ; c.psittaci test by PCR in the lesionnal tissue and the conjunctival swab; search for Gougerot-Sjögren syndrome.
2- … is therefore often not recommended [80] Correct puntuation to ensure.
3- Figure 4. The word immunotherapy is cut.
4- Consider the role of radical surgery in the treatment of localized tumors such as the conjunctiva and the lacrimal glands and question the problem of the adjuvant treatment in case of incomplete excision.
5- The effectiveness of the anti-lymphoma efficacy of antibiotic therapy against Chlamydia psittaci is discussed in ocular adnexa lymphoma localization.

6.4. Other EMZL

1- POD24: meaning.
2- If this is not possible, treatment should be based on several factors including: type of previous therapy, time to relapse from previous treatment, disease status and performance status.
Add : in relapsed extranodal MZL, usually systemic therapy is needed (single-agent rituximab, chemo-immunotherapy, single-agent BTK-inhibitor…).

Author Response

Dear Revisor 2,

Thank you for your comments, we  tried to address all your suggestions, surely improving of the work. Find the paper revised attacched.

Here you will find point by point answer to your comments. All the modifications in the text addressed to your specific observations are highlighted in yellow

In order to group the message:

MZL accounts for approximately 10% of non-Hodgkin's lymphomas with a mean age greater than 60 years at diagnosis. In the United States, the incidence is 18.3 cases per 1 million person-years [2].

Thank you, We added the sentence to complete epidemiology section.

1- Emphasize the importance of the presence of complete or partial trisomies of the long arm of chromosomes 3 and 18 (+3/+3q or +18/+18q) which are common to all subgroups of LZM but their association is important for differentiate LZMs from other SLP-Bs.

2- Cytogenetic abnormalities could be brought together in a specific paragraph. A trisomy of chromosome 3 or 8 is highly suggestive of marginal zone lymphoma but is not specific. Other non-specific anomalies to be specified in this paragraph such as gain of 6p25 and ocular adnexa EZML….

3- Specify right from the start in this paragraph that the incidence of translocations varies according to the anatomical site: t(1;14)(p22;q32) inducing BCL10-IGH is found more in the pulmonary and intestinal MALTs; the t(14;18)(q32;q21) inducing IGH-MALT1 is found more in ocular lymphomas then of the salivary glands, the t(3;14)(p14;q32) inducing FOXP1-IGH is found mainly in those of the thyroid then in the MALT of the orbit and the skin.

Express the molecular fusion in agreement with the carrier chromosome.

Your observations helped to improve the paragraph. Etiopathogenesis paragraph was entirely revisited.

« Once the underlying infection or autoimmune disorder is detected, specific treatment is as well responsible for tumor remission ». Consider : …., although antibiotic therapy may have limits in its anti-lymphoma potential in extragastric mucosal sites. Zucca E, Arcaini L, Buske C, et al. Marginal zone lymphomas: ESMO clinical practice guidelines for diagnosis, treatment and follow-up. Ann Oncol 2020;31:17-29. Marginal-zone lymphomas. Rossi D, Bertoni F, Zucca E. N Engl J Med. 2022.

Thank you, Sentence was modified and references added (9 and 26)

  1. Clinical presentation and disease assessment

1- Lung EMZL: Recurrent respiratory infections and/or discovery of lung nodules on imaging studies.

2- Ocular EMZL : slow growing masses, eye redness.

Thank you for your observations. Proper changes were made

  1. Pronostic factors and disease features

1- The MALT International Pronostic Index (IPI)

Thank you for your observations. Proper change was made

4.1. EMZL of gastrointestinal tract or gastric MALT
A- Consider the following remarks:
1- Decrease in the incidence of gastric MALT lymphoma with Helicobacter pylori eradication.
2- « It has nowadays commonly consolidated the model in which chronic antigenic exposure and inflammation sustain the development of the disease ». Delete : … is consolidated
3- …. T-regulatoty cells and cytokines … are involved in the inflammation creating the lymphoma pathogenesis together with…
4- « Sometimes immunoglobulins are detectable in false positive immunofixation [23] ». : not clear.
5- ….a high percentage of blasts : replace with « centroblasts ».
6- « Anemia caused by chronic gastric bleeding » : replace with « Iron deficiency anemia ».
7- Mucosal biopsy performed during endoscopic procedure is the gold standard for diagnosis.

- Consider lymphoepithelial lesions in the anatomo-pathology analysis. In mucosal tissues, the clonal marginal zone B cells typically infiltrate the epithelium, forming lymphoepithelial lesions which are considered as a hallmark of MALT lymphoma, although not indispensable for diagnosis.
- Consider the demonstration of monoclonality in order to rule out the diagnosis of inflammation and to confirm lymphoma.

Your remarks were considered and proper changes performed. Especially we solved some understanding issues with anatomo-pathology analysis and monoclonality.

8- Consider endoscopy ultrasonography if possible, in gastric wall infiltration assessment and/or perigastric lymph nodes.
B- « MALT cells are similar to plasma cells » : not clear
C- « Around 25 to 30% of gastric MALT are estimated harboring the BIRC3-MALT1 translocation which guarantees a survival advantage to tumor cells and is absent in pulmonary or ocular adnexa MALTs ».
Yet t(11;18) (q21;q21) / BIRC3-MALT1 can be found in about 40% of pulmonary forms.

In accordance with the statement in paragraph 4.2: “This translocation is often detected in the lung and gastric EZML »

Your remarks were considered and proper changes performed. Especially we solved some understanding issues with anatomo-pathology analysis and monoclonality.

4.2 Bronchus-associated lymphoid tissue (BALT) lymphoma
1- In staging patients with primary involvement of the lungs, or upper airwairs, or Hp infection, consider an endoscopy of the upper digestive tract for lymphoma silent localization.

Consideration added

4.3. Ocular adrexal marginal zone lymphoma (OAML)
Potentially impaired ocular motility. Is not a sentence.

Sentence corrected

4.4. Other EMZL sites

ENT examination and echography and EGD. Specify the meaning of the abbreviations. ENT: ear, nose and throat ? EGD: oesophagastroduodenoscopy ?

Abbreviations outlined.

Treatment of EMZL

1- In all primary EMLZ sites, consider the initial staging with clinical evaluation, hemogram and cytological blood analysis ; biochemical, renal and liver tests, with lactate deshydrogenase ; serum (urine) protein electrophoresis ; HBV, HIV and HCV serologic tests ; possibly completed by a PET-CT for the purpose of assisting in medical decision-making, in biopsy site, staging, fear of histological transformation; bone marrow aspirate and biopsy in case of doubt about bone marrow involvement (cytopenia, bone hyperfixation in PET).

Staging section is reported above  in respective paragraph for each subtype

2- « It can be used in patients with modest residual disease after surgery, RT or antibiothic therapy », add : nevertheless considering the question of adjuvant treatment.

Sentence added

1- To specific Noxa such as.. Noxa meaning ?

Word changed

2- Recently, Bruton-Kinase inhibitors (BTK), such as Ibrutinib, have been tested on MZL, showing efficacy on previously treated patients (n 30): Duration of response and PFS at 33 months were 50% and 26% respectively [66] Correct punctuation to ensure.

Punctuation corrected

Also, consider the following data:

1- If the immunohistochemical study for the search for Helicobacter pylori is negative, carry out a fecal antigen or a breath test and a serological study.

2- Prefer anti-infective treatment to gastrectomy with potential sequelae, in Hp+ gastric lymphoma.

3- A second line anti-Hp treatment may be necessary in less than a quarter of cases.

4- In localized Hp+ gastric lymphoma, only an eradicating anti-infectious treatment is put in place, with a study of the histological response which can be slow.

5- Mucosal monoclonal B lymphocytes may persist despite histological regression.

6- Gastric lymphomas with chromosomal translocation t(11;18)(q21;q21) (detection by in situ hybridization FISH or PCR) have a tendency to tumoral progression and show a poor response to antibiotic therapy, even in the presence of the infectious agent. In this case, consider anti-infectious treatment secondarily associated with localized radiotherapy.

7- The anti-infectious treatment will probably have little or no effect in the event of negative detection of Helicobacter pylori associated with the MALT lymphoma.

8- Localized radiotherapy should be considered in the context of localized gastric MALT lymphoma that does not respond to antibiotic therapy or negative for Hp.

Thank you for your helpful observations. We tried to globally improve information inside this paragraph

6.2. BALT lymphoma

Consider the BALT lymphoma initial staging with bronchoscopy and bronchoalveolar lavage (BAL); possibly completed by a PET-CT and esophagogastroduenoscopy.

Staging section is reported above in respective paragraph for each subtype

1- Figure 3 : The word immunotherapy is regularly cut.

2- Advanced patietns : change to « Advanced patients ».

Changes performed

6.3. Treatment : OAML
1- Consider the OAML initial staging with ophtalmologic clinical evaluation, orbit and head and neck magneric resonance imaging (MRI) ; possibly completed by a PET-CT ; c.psittaci test by PCR in the lesionnal tissue and the conjunctival swab; search for Gougerot-Sjögren syndrome.

Staging section is reported above  in respective paragraph for each subtype

2- … is therefore often not recommended [80] Correct puntuation to ensure.
3- Figure 4. The word immunotherapy is cut.
4- Consider the role of radical surgery in the treatment of localized tumors such as the conjunctiva and the lacrimal glands and question the problem of the adjuvant treatment in case of incomplete excision.
5- The effectiveness of the anti-lymphoma efficacy of antibiotic therapy against Chlamydia psittaci is discussed in ocular adnexa lymphoma localization.

Figure 4 was edited, punctuation corrected and radical surgery role was better highlighted

6.4. Other EMZL

1- POD24: meaning.

Meaning added

2- If this is not possible, treatment should be based on several factors including: type of previous therapy, time to relapse from previous treatment, disease status and performance status.

Add : in relapsed extranodal MZL, usually systemic therapy is needed (single-agent rituximab, chemo-immunotherapy, single-agent BTK-inhibitor…).

Sentence added

Round 2

Reviewer 1 Report

Thank you for a substantial revision of the manuscript.

Most of the points were sufficiently addressed. However, some minor inconsistencies remain:

Please review again the designations of chromosomal translocations throughout the article. There is a "t" missing in section 2.1, and there is still some inappropriate punctuation with spaces and commas, especially in the second part of the paper.

Also in 2.1 translocations are described as mutations. While in a broad sense it is true, the use of the term "mutation" is usually reserved for small, nucleotide-level changes. Just replace "mutations" with "translocations"

The same can be said about the designation of microbial species. In section 5. there is still the designation "Cj", probably meaning C. jejunii. Please check this all across the manuscript.

In section 4.3 authors write that "in 55% of cases, the PCR analysis shows the rearrangement of the heavy chain of immunoglobulin and somatic hypermutations in 60%". Is IGH rearrangement or class switch of the heavy chain is meant? Given that the MZL originates from post GC germinal cells, one would expect that all cases have rearranged Ig heavy chain.

At the end of the same paragraph, there is a reference to "the MYD88  mutation". Probably for most readers, it will be clear that the L256P mutation is meant, but it wouldn't harm to state it precisely.

Section 4.4 ends with "and so on". This is not helpful for the reader. Authors should decide if it is an important information and not. If so, then a full description is warranted. If not, then the entire part can be left out.

Figure 2 actually shows the management flowchart of the H. pylori-positive gastric MZL. The title of the figure has to be supplemented accordingly, or a figure should be expanded to contain management options for a H. pylori-negative gastric MZL.

Last but not least, I feel that the description of the genetic background of different types of EMZL has not been sufficiently addressed. Especially some recent important publications on somatic mutations beyond chromosomal abnormalities have been largely ignored (e.g. PMIDs:  29765142 34494161, 32012328, 27566587 28152507, 32316399).

Author Response

Dear Reviewer 1,

Thank you again for your comments, here you find a point to point answer to your last observations.

The cytogenetic nomenclature has been revised along the whole text; also the confusion between mutation and translocation has been clarified (in yellow).

The designation Campylobacter Jejuni, where appropriate, has replaced Cj.

In section 4.3 the aspect of IGH rearrangement and somatic hypermutation has been clarified (in yellow).

The exact designation of L256P MYD88 mutation has been specified (Yellow).

The term "so on" was eliminated from the text.

According to your suggestion, figure 2 was modified introducing a new section concerning H.pylori-negative gastric Malt.

According to you suggestion, specific paragraphs were implemented with more extensive genetic background in different EMZL subtypes. (see paragraph 4.1 to 4.3 in yellow)